# Physiological Characteristics of Sunburn Peel after Apple Debagged

**DOI:** 10.3390/molecules27123775

**Published:** 2022-06-11

**Authors:** Yifeng Feng, Shanshan Li, Rongjian Jia, Jie Yang, Qiufang Su, Zhengyang Zhao

**Affiliations:** College of Horticulture, Northwest A&F University, Xianyang 712100, China; 15909073303m@sina.cn (Y.F.); eason_1216_5360@163.com (S.L.); jrj16882021@163.com (R.J.); y-j@nwsuaf.edu.cn (J.Y.); qfsu@nwafu.edu.cn (Q.S.)

**Keywords:** *Malus domestica* Borkh., ‘Gala’, debagged, non-bagged, sunburn

## Abstract

The bagging of fruits provides efficient protection from high-intensity sunlight and improves fruit color and quality. However, bagged fruit suddenly exposed to bright light can cause sunburn and destroys the peel cell structure. In this study, fruits from ten-year-old apple trees of ‘Gala’ variety were debagged, and the effect of sunburn on fruits was divided into: (1) normal peels (BFN), (2) peels with albefaction (BFA), and (3) browning (BFB). The non-bagging fruits (NBF) were set as a control to study the physiological characteristics of apple fruits with different levels of sunburn. Our results showed that in the early stages of debagged fruits’ sunburn, the cell structure of the peel was partially destroyed, the color of the injured fruit surface turned white, and the peroxidation in the cell membrane of the peel increased. Initially, the fruit improved its photosynthetic protection ability, and the activity of antioxidants and phenolics was enhanced, to cope with external injury. However, with the increase in duration of high-intensity sunlight, the cell structure of the peel was severely damaged, and the increase in membrane peroxidation resulted in brown coloration of fruits. Under the same conditions, the photoprotection ability and antioxidant enzyme activity of non-bagged fruits showed higher levels. In conclusion, the non-bagged fruits were more adaptable to high-intensity sunlight as compared to debagged fruits.

## 1. Introduction

Apple (*Malus domestica* Borkh.) is one of the major fruits growing throughout the world [1]. Bagging of the apple fruits is a major cultivation measure to effectively improve the quality of apple by having a clean fruit surface, better color, and no or minimum pesticide residue [2,3]. However, fruits that have been bagged for a long time are very sensitive to sunlight, and the direct solar radiation or reflected light often leads to photooxidative sunburn, and its extended exposure results in sunburn [4]. There are three types of sunburn on apple: sunburn necrosis, sunburn browning, and photooxidative sunburn [5].

The damage caused by sunburn causes discoloration of the peel and a higher respiration rate, which leads to degradation of anthocyanins and chlorophyll contents [6]. The measurements of chlorophyll fluorescence can reflect the degree of plant damage [7]. The higher light intensity and temperature damages photosystem II (PSII), and high temperatures significantly reduce the photochemical activity of PSII primary light energy conversion efficiency *(F_v_/F_m_*) [8,9].

Under stress conditions, plant metabolism is disturbed by over-production of reactive oxygen species (ROS) [10,11]. The over-produced ROS can damage the lipids, proteins, and DNA of peel cells, ultimately leading to oxidative stress in plants [12]. To avoid the oxidative stress caused by the excessive accumulation of ROS, the plants correspondingly form a comprehensive defensive system composed of enzymatic and non-enzymatic antioxidants to reduce damage and maintain healthy growth [5]. According to Zhang et al. [6], the activities of superoxide dismutase (SOD) and catalase (CAT) were higher in sunburned fruits in apple. CAT catalyzes the breakdown of hydrogen peroxide in cells and its activity is highly dependent upon the metabolic rate and resistance to disease or oxidative stress [13]. SOD and POD have high infinity for ROS and eliminate hydrogen peroxide in plants [14]. The activity of polyphenol oxidase (PPO) is higher in fruits with a physiological disorder or during the damaging of tissues in plants. The substrate of PPO is activated and reacts with oxygen to produce quinone, which can also regulate the rate of harmful photooxidative reaction in chloroplasts and enhance plant resistance [15]. Moreover, anthocyanins, carotenoids, and phenols also play a major role in scavenging excessive hydrogen peroxide produced in response to environmental stress [16]. Carotenoids absorb light energy and transfer of light energy in a molecular state that does not participate in light absorption transfer, preventing destructive photooxidation [17]. Proline indirectly reflects the resistance of plants under adverse conditions, by forming a protective film with water molecules to maintain cell homeostasis [18,19,20]. Moreover, phenolic compounds have a strong UV absorption capacity between 200 and 300 nm, which also plays an important role in protecting apples from UV-B radiation damage [21].

Sunburn is a physiological disease that occurs in fruits exposed to excessive solar radiation [6], which seriously affects the quality of the fruit, and it also occurs in fruits other than apples [5]. Sunburn has caused huge losses in several apple-producing regions in the world [21], for example, in Washington State, the main apple-producing region in the United States, where sunburn is one of the main factors affecting local apple production, with an average loss of about 10% per year [5].

The Qingcheng County, Gansu Province, is arid and dry all year round. The highest light intensity and temperature of the year are usually recorded in the month of August. Therefore, such climatic conditions are very conducive to the occurrence of apple sunburn. Due to the high altitude and dry climate, sunburn is a major factor affecting the development of the local apple industry, which is relatively common. Several studies have determined the role of fruit bagging in improving apple fruit quality and yield [3,5]; however, the debagging of fruits prior to harvesting leaves them prone to sunburn, and the differences in physiological index changes between debagged and non-bagged apple fruits remain to be further studied. Very little information is available in the literature comparing debagged and non-bagged fruit sunburn. Therefore, in our work, ‘Apple Cv Gala’ was used as the test material to study the physiological indexes of apple fruits under different levels of sunburn in debagged and non-bagged apple fruits to further reveal the potential causes of apple sunburn formation.

## 2. Results

### 2.1. The Influence of Environmental Factors on Sunburn

We investigated the atmospheric temperature, humidity, light intensity, and fruit surface temperature of five sunny days during the sunburn period (Figure 1). At our experimental station, the variation trend of light intensity and air temperature was consistent, and both of the maximum values were reached at 14:00 p.m. (Figure 1a,c). However, the highest surface temperature of apple fruit (44.2 °C) was relatively late in the afternoon, i.e., around 16:00 pm (Figure 1b), which was 7.8 °C higher than the maximum temperature of the air. Air humidity was the lowest at 14:00 pm, only 20% (Figure 1d), which was contrary to the trend of orchard temperature (Figure 1c).

### 2.2. Effect of Sunburn on Cell Microstructure

The scanning electron microscopy of fruit peel in our work is presented in Figure 2. The anatomical structure of BFN and NBF fruit peels showed a smooth cell surface and orderly arrangement of cells with no damage (Figure 2a,b). The structures of BFA peel cells were slightly deformed, with rough surfaces, and the spaces between cells were larger (Figure 2c). BFB peel cells were obviously deformed and atrophied, the cells’ arrangement was irregular, and the epidermal cells were ruptured. The number of ruptured cells was higher, and the peel surface was fractured (Figure 2d).

### 2.3. Effect of Sunburn on Peroxidation of Pericarp Membrane Lipid

In our work, there was no significant difference in O_2_^−^ content between BFN and BFA and NBF (Figure 3a). The concentration of O_2_^−^ content in BFB was significantly higher than the others, demonstrating 22% higher than BFN (Figure 3a). The MDA content of BFB was also significantly higher than the others (Figure 3b). The content of O_2_^−^ and MDA in NBF was low (Figure 3), reaching 87% and 48% of that of BFB, respectively.

### 2.4. The Change of Chlorophyll Fluorescence Parameters

As shown in Figure 4, the minimum *F_v_/F_m_* value was observed in BFB fruits, which was 59% of that of BFN, and there were no significant differences between NBF and BFN (Figure 4a). The increased injury levels decreased both Y (I) (the actual PSI efficiency) and Y (II) (the actual PSII efficiency), and vice versa, and the light energy conversion efficiency of NBF was similar to that of BFA (Figure 4b,c). Y (NO) (the quantum yields of unregulated energy dissipation at PSII) changed little upon debagging, and Y (NO) of NBF was the highest (Figure 4d). The Y (NPQ) (the quantum yields of regulated energy dissipation at PSII) value of BFA was 28% higher than that of BFN, while the Y (NPQ) value of BFB was the lowest (Figure 4e). In early stages of sunburn, the NPQ (the non-photochemical quenching coefficient) value increased and decreased rapidly with the aggravation of epidermal damage (Figure 4f). The photosynthetic fluorescence parameters of NBF remained at a middle level overall (Figure 4).

### 2.5. Comparison of Antioxidant Enzyme Activities

In bagged fruits, the contents of CAT, PPO, and APX in BFB were 1.57, 0.98, and 2.51 times those in BFN, which were significantly higher than other treatments (Figure 5a,c,e), and this indicates that the three enzymes were persistent in resisting sunburn stress of apple fruit. The POD and SOD values of BFA were the highest (Figure 5b,d), which were significantly higher than those of BFN and BFB, and this indicates that these two enzymes played little role in resisting sunburn stress at the later stage. Besides, most of the NBF fruits have strong enzymatic activity (Figure 5).

### 2.6. Pigment Content of Peel

The chlorophyll content of NBF was significantly higher than that of non-bagging fruits, reaching 18.07 ug g^−1^, which was significantly higher than that of the other three groups. In debagged fruit, the lowest chlorophyll content was observed in BFB, which was 35.77% lower than BFN (Figure 6a). In this study, the content of TMA in BFN was the highest and BFB was the lowest in debagged fruits. The content of TMA in NBF was the lowest as compared to the other three groups (Figure 6b).

### 2.7. Levels of Phenolic Compounds in the Peel

In our work, a total of 17 phenolics were detected in ‘Gala’ peels by HPLC-MS (Table 1). The rutin, chlorogenic acid, and isoquercitrin accounted for more than 85% of the total phenolic compounds content, procyanidin B2 and epicatechin accounted for more than 10% of the total phenolic compounds content, while ferulic acid, methyl gallate, p-hydroxycinnamic acid, caffeic acid, and their contents accounted for less than 1% of the total phenolic compounds (Table 1). We observed that the content of phenolic substances in BFA was significantly higher than the other three treatments in fruit peels (Table 1). The content of phenols in BFN was the lowest, 0.39 mg g^−1^ lower than that of non-bagging fruits (NBF).

## 3. Discussion

For the debagged fruit, it is easy to form sunburn under high temperature and strong light, and it was often observed that the peak of fruit sunburn always coincides with the highest temperature period of the year in production [5]. Higher temperatures and increases in sunlight intensity have a significant role in degrading the quality of fruits. The higher environmental temperature causes the fruit surface to cross its threshold level, and may lead to sunburn [22,23]. In our work, the maximum light intensity and air temperature were observed at the same time, indicating that sunburn could be due to the combination of heat and light [5,24]. The maximum temperature of the fruit surface was relatively delayed, reaching a maximum of 44.2 °C at 16:00 p.m. (Figure 1b), which was 7.8 °C higher than the maximum temperature of the air. This showed that when the light intensity and air temperature were highest, the fruit surface temperature continued to accumulate and part of the light energy was converted into heat energy, thereby increasing the surface temperature of the fruit [5]. Low air humidity can aggravate sunburn, as high air humidity facilitates the 5‘-nucleotidase activity and reduces sunburn damage [5]. In our work, air humidity dropped to the lowest at 14:00 p.m., indicating that air humidity also affects sunburn. Combining temperature, light intensity, and air humidity, we found that 14:00 to 16:00 pm is the critical time for the formation of ‘Gala’ apple sunburn occurrence.

Sunburn in fruits triggers the production of ROS, and their over-production leads to cell membrane peroxidation, extravasation of intracellular fluid, and an imbalance in metabolism. The over-production of O_2_^−^ (the intermediate product of membrane lipid peroxidation) leads to oxidation in cells and causes degradation of lipids, which increases the MDA content in cells [3,25,26,27]. In our study, the O_2_^−^ content of BFN and BFA showed no significant differences, which indicates that at the beginning of sunburn injury, the higher scavenging effect in the pericarp cells overcomes the increasing concentrations of ROS [28]. The O_2_^−^ content of BFB was significantly higher than that of BFN and BFA, which can be corelated to the fact that with increasing stress conditions, the ability of cells to scavenge ROS decreases and results in cell membrane degradation [26,29,30].

Chlorophyll fluorescence parameters can reflect the photosynthetic mechanism and physiological status of plants and indirectly reflect the health status, which is regarded as a probe to study the relationship between plant photosynthesis and the environment [31]. *Fv/Fm* reflects the potential maximum photosynthetic capacity of plants and can effectively indicate the levels of plant stress [32]. For plants, upon encountering adversity, the maximum photosynthetic efficiency of *Fv/Fm* will decrease [7], which can be corelated to our results as the *Fv*/*Fm* of debagged fruits decreased with increasing sunburn. In debagged fruit, the higher Y (NPQ) value of BFA indicates that the photoprotection ability of pericarpium was enhanced at the beginning of injury, while the lower Y (NPQ) value of BFB reveals that the photoprotection effect decreased with the further increase of injury [32]. The higher photosynthetic fluorescence parameters of NBF indicate that the photoprotection ability of non-bagged fruits was stronger than that of debagged fruits. In shaded fruits, the PSII reaction center would make the peel more vulnerable to excess light energy under bright light, and part of the PSII reaction centers will be reversibly deactivated and become an energy trap, which cannot transmit the absorbed light energy to the electron transport chain [5]. This also explains the sensitivity of debagged apple fruits to sunburn.

The biochemical characteristics of different antioxidant enzymes make them have different inducibility in gene expression levels, which enables the antioxidant system to scavenge excessive ROS [6]. CAT, POD, PPO, SOD, and APX are the main antioxidant enzymes for scavenging reactive oxygen species (ROS) [3,33]. SOD is the first line of defense against oxidation in plants. It scavenges excessive O_2_^−^ produced in cells and results in H_2_O_2_ and O_2_ as byproducts, while CAT, POD, and APX remove excessive H_2_O_2_ from the cells [34]. However, with the increase in stress, cells and the inner membrane of organelles are damaged. Under such circumstances, PPO oxidizes endogenous phenolic substances, catalyzes the generation of quinones, and leads to the appearance of brown spots on the peel [3]. In our work, we observed that BFB had the highest PPO content, which can be corelated to the appearance of brown spots on BFB.

In our work, we noted that the peels of apple turned white in the early stages and brown in severe burns. This can be attributed to a loss of chlorophyll and anthocyanins in the peel cells [6]. In production practice, the unbalanced solar radiation may cause the photosynthetic pigment of fruits to be destroyed [5]. In recent years, there are numerous studies showing anthocyanin’s role in plant resistance to stress conditions. According to Bi et al. [35], O_2_^−^ scavenging activity in apple peel showed that anthocyanin was more sensitive to H_2_O_2_ and contributed more to H_2_O_2_ scavenging than other phenolic compounds. Studies have shown that anthocyanins are also involved in protecting fruits from ultraviolet radiation and excessive light energy [11]. Our findings suggest that the chlorophyll content of ‘Gala’ decreased rapidly after debagging and decreased faster with the deepening of injury levels. The content of anthocyanin in BFB treatment was higher than that in NBF, which may be due to the continuous accumulation of anthocyanins in brown peel [35].

It is well-known that phenolic compounds are important secondary metabolites that play an important physiological role throughout the lifecycle of plants, and their antioxidant capacity has been studied for a long time for their ability to scavenge harmful reactive oxygen species [36]. Oxidative stress caused by abiotic stresses such as drought, strong light, UV exposure, and temperature often leads to the accumulation of phenolic compounds [37]. In our work, in the debagged treatment, phenolic compounds in BFA were higher than in BFN, and lower than in BFB, and this indicated that the higher phenolic compounds at minor injury levels were due to active defense mechanisms and decreased with increasing injury levels due to the irreversible damage to the peel cells [38,39]. Similar to other studies, stress only increased the level of phenolic compounds to a certain extent, and severe stress may inhibit the accumulation of phenolic compounds [40]. The lower content of phenols in NBF suggests that non-bagged apples may adapt to high-temperature and high-light environments, which inhibited the injury caused by the stress, consistent with the previous reports of Wang et al. [3] and Sun et al. [41]. In this study, rutin and chlorogenic acid accounted for the largest proportion among the phenolic compounds, indicating that these two substances played an obvious role in resisting burning stress of apple. It has been reported that rutin, as a natural flavone derivative, plays a protective role in oxidative stress by combining with metal chelation and free radical-scavenging activities [42], and it can induce oxidative damage membrane components, thus enhancing cell tolerance and reducing oxidative stress [43]. Chlorogenic acid (CGA) is the first phenolic product produced by the phenylpropanoid biosynthetic pathway, which originates from higher plants under biotic or abiotic stress [44]. Mei and Sun found that exogenous CGA has the effect of scavenging free radicals, which can reduce electrolyte leakage in apple leaves, and exogenous CGA is involved in protecting apple leaves from oxidative stress [40]. We are increasingly convinced that the accumulation of phenolic compounds can reinforce the resistance of the system to oxidative damage.

## 4. Materials and Methods

### 4.1. Experimental Design

The test apples, ‘Gala’ variety, came from Qingcheng Apple research station of Northwest Agriculture and Forestry University in China (36°0′14″ N, 107°54′43″ E). The test station has an altitude of 915 m, a temperate continental monsoon climate, an average annual rainfall of 537.5 mm, an average annual temperature of 9.4 °C, and a frost-free period of 150 days. The incidence of apple sunburn was about 15% in the orchard after debagging. The trees selected were 10 years old, grafted on M.26 rootstock, and were planted at a spacing of 2 × 4 m. After 30 days of blooming, 10 apple trees were randomly selected on 15 May 2021 for covering the fruits and another 10 trees as a control, where their fruits were not covered in bags. The bags used were double-opaque, with an inner layer of red and an outer layer of brown color. All trees were grown using standard horticultural practices with recommended disease and pest control measures.

The bags around the fruits were removed from the debagged apples group after 75 days (30 July 2021). The small twigs and leaves around the fruits were removed from both debagged and non-bagged fruits to achieve full exposure to sunlight. After 15 days, different degrees of sunburn appeared on the debagged fruits: no sunburn (normal), more than 1/5 of the sunny side of the peel showed white discoloration (albefaction), and more than 1/5 of the sunny side of the peel showed brown discoloration (browning), and we find that there was no sunburn in non-bagged fruit (Figure 7). The debagged fruits with normal (BFN), debagged fruits with albefaction (BFA), and debagged fruits with browning (BFB) were randomly selected as the experimental groups, while non-bagged fruits (NBF) were used as the control. Thirty fruits of similar size and location were selected from each group, and each group was treated with three biological replicates, with ten fruits per replicate. A 0.2 mm peel was sampled by cutting with a sterile scalpel from partial sunburn (albefaction) and severe sunburn (browning). The peels from normal fruits were also randomly sampled from the sun-facing surface of the fruits. The collected samples were immediately frozen in liquid nitrogen and stored at −80 °C for further analysis, in addition to the determination of chlorophyll fluorescence and the cell microstructure indicators.

### 4.2. Environmental Factors and Fruit Surface Temperature

Sunlight intensity for five random sunny days was recorded from debagging to harvesting with a high-precision illuminance meter (TES-1339R). Similarly, the temperature and humidity of the orchard were measured with an electronic precision long-time thermo-hygrograph (L95-2 LUGE). Moreover, the part of the fruit exposed to temperature was measured with an infrared high-precision handheld thermometer (Aicevoos AS-D400). The recordings were noted every hour from 08:00 am to 18:00 pm, with 5 biological repeats.

### 4.3. The Parameters of Chlorophyll Fluorescence

A Dual PAM-100 Chlorophyll Fluorometer (Heinz Walz GmbH, Nuremberg, Germany) was used to determine peel chlorophyll fluorescence measurements after dark adoption for about 30 min, as described by Klughammer et al. [45] with the following modifications. A 1.5 cm diameter of light-affected fruit peel (with some pulp) was sampled from 10 fruits, and this procedure was repeated 5 times.

The chlorophyll fluorescence measurements were calculated as follows [7,8]:(1)$$Fv/Fm=\frac{Fm-Fo}{Fm}$$
(2)$$Y\left(\mathrm{NO}\right)=\frac{F}{Fm}$$
(3)$$Y\left(\mathrm{NPQ}\right)=\frac{F}{Fm^{\prime }}-\frac{F}{Fm}$$
(4)Y(II)=1−(Y(NO)+Y(NPQ))
(5)$$\mathrm{NPQ}=\frac{Y\left(\mathrm{NPQ}\right)}{Y\left(\mathrm{NO}\right)}$$
where *Fv*/*Fm* is the PSII potential quantum efficiency, Y(NO) and Y(NPQ) are the quantum yields of PSII unregulated and regulated energy dissipation, respectively, Y(II) is the actual PSII efficiency, NPQ is the non-photochemical quenching coefficient, and *Fm’* and *Fm* are the maximum fluorescence values determined after light adaptation and dark adaptation for 30 min, respectively.
(6)$$Y\left(I\right)=\frac{Pm^{\prime }-P}{Pm}$$
where Y(I) is the actual PSI efficiency, *P* is the P700 signal in the light, and *Pm’* and *Pm* are the maximum P700 signals measured using saturated pulses after far-red light illumination during light adaptation and dark adaptation, respectively.

### 4.4. The Microstructure of Peel Cells

The microstructure of peels cells was determined as described by Wang et al. [3], with the following modifications. Briefly, the fruit peel (5 × 3 × 3 mm) was fixed with 4% glutaraldehyde fixative for 6 h, followed by 10%, 30%, 50%, 70%, 80%, 90%, 95%, and 100% ethyl alcohol, and 25%, 50%, 75%, and 100% tert-butyl for 15 min each, respectively. Afterwards, the sample was freeze-dried for about 3 h, glued, gilded, and observed under a S-3400N Electron Microscope (acceleration voltage 5–15 kV).

### 4.5. The Activities of Enzymes

The sample preparation for assaying the activities of antioxidant enzymes was performed according to Lo’ay [46], with the following modifications. First, 0.5 g of fresh peel was ground in the presence of 5 mL of 0.05 mol L^−1^ phosphate buffer (pH = 7.8), with 1% (*w/v*) polyvinylpyrrolidone (PVP). Afterwards, the homogenate was centrifuged at 4 °C, at 13,000 rpm for 20 min. The obtained supernatant was used for assaying the activities of the following antioxidant enzymes.

The catalase (CAT) activity was determined by measuring H_2_O_2_ reduction according to Zhang et al. [47], and the peroxidase (POD) activity was determined through the guaiacol method by Tang et al. [48]. The polyphenol oxidase (PPO) activity was determined following Lin et al. [49]. The total superoxide dismutase (SOD) activity was determined by the inhibition of the photochemical reduction of nitroblue tetrazolium (NBT) generated by superoxide radicals [50], and the ascorbate peroxidase (APX) activity was measured according to Ahn et al. [51].

### 4.6. O_2_^−^ and MDA

O_2_^−^ was determined according to the method described by Lo’ay et al. [46], with some minor modifications. The fruit peel (0.5 g) was homogenized in 5.0 mL of 50 mmol L^−1^ of phosphate buffer (pH 7.8) and mixed. Then, 1.0 mL of supernatant was mixed with 1.0 mL of 1 mmol L^−1^ of hydroxylamine hydrochloride, 2.0 mL of 17 mmol L^−1^ of p-aminobenzene sulfonic acid, and 2.0 mL of 7 mmol L^−1^ of α-naphthylamine. The absorbance was measured by a spectrophotometer (UV-1800, Shanghai, China) at 530 nm.

The MDA content was measured through the thiobarbituric acid (TBA) reaction method [10]. The 1.0 mL supernatant obtained was added to 2.0 mL of 0.67% thiobarbituric acid (TBA) dissolved in 5% (*v/v*) trichloroacetic acid (TCA). The mixture was heated in a water bath for 15 min and cooled to room temperature. The extract was than centrifuged at 500 rpm for 20 min, at 4 °C, to allow the precipitate to settle down. The supernatant was carefully selected and used for the spectrophotometer (UV-1800, Shanghai, China) determination of MDA at 450, 532, and 600 nm wavelengths.

### 4.7. Chlorophyll and Total Monomeric Anthocyanin

Total chlorophylls in fruit peel were measured by extracting 0.5 g of peel tissue in 5 mL of ethanol (95% *v/v*). The samples were kept at 4 °C for 24 h. The extract was centrifuged at 10,000 rpm for 10 min. The absorbance of the extract was observed on a spectrophotometer (UV-1800, Shanghai, China) at 665 and 663 nm wavelengths [52].

Total monomeric anthocyanin in peel tissues was measured according to Ahmed et al. [53]. The fruit peel of 0.5 g weight was ground in liquid nitrogen, and 10 mL of hydrochloric acid/methanol solution (*v/v* = 1:99) was added. The samples were placed in a shaker under a dark condition for 24 h. The supernatant was used for spectrophotometer (UV-1800, Shimadzu, Japan) determination at 530 and 657 nm.

### 4.8. Phenolic Compounds

Phenolic compounds in peel tissues were measured according to Zhang et al. [54] (with minor modifications), with three biological repetitions. Briefly, 0.2 g of fruit peel was added to 2.0 mL of methanol and water (*v/v* = 8:2), and sonicated at 40 °C for 60 min. The samples were centrifuged at 5000 rpm at 25 °C for 15 min. The obtained supernatant was passed through a 0.22 μm organic phase filter membrane and stored in small brown tubes for further analysis with high-performance liquid chromatography-mass spectrometry (HPLC-MS/MS, AB QTRAP 5500, AB SCIEX corporation of USA).

Liquid chromatography analysis was performed on an AB SCIEX instrument. An InertSustain AQ-C18 (150 × 4.6 mm, 5 µm; GL Sciences) was used for chromatographic separation (column temperature was 35 °C). The sample injection volume was 5 μL, and the flow rate was 0.7 mL/min of a binary mobile phase (A: aqueous solution with 1% formic acid, B: acetonitrile). The percentage of A was as follows: 0–1 min, 100–75%; 1–5 min, 75–5%; 5–6.5 min, 5%; 6.5–6.5 min, 5–75%; 6.6–8 min, 75%; 8–10 min, 100%. The standards’ concentration was 10–400 ng/ mL.

Mass spectrometric analysis was performed using an AB SCIEX ion-trap mass spectrometer QTRAP 5500, operated in negative electrospray ionization (ESI) mode. Multiple reaction monitoring (MRM) mode was applied to detect phenolic compounds, with a 50 ms dwell time for each transition.

Quantification of phenolic compounds was achieved using the ratio between the area under the peaks for the analytes and comparison with standard curves (Appendix A). All the phenolic standards were obtained from Yuanye Bio-Technology Co., Ltd. (Shanghai, China).

### 4.9. Statistical Analysis

Experimental treatments were arranged in a completely randomized design. Data were subjected to analysis through one-way ANOVA using SPSS software (version 16.0, SPSS), and means were separated for significance by using Duncan’s multiple range test. Graphs were created through the Microsoft Excel 2016 built-in function for making graphs, and the *ggplot2* function in R studio (3.6.1 version) [55]. The results were expressed as mean ± SE. If the resulting *p*-value was lower than 0.05, the difference was considered significant.

## 5. Conclusions

In conclusion, in debagged fruits, at the early stage of sunburn, photosynthetic protection ability, antioxidant enzyme activity, and protective substances of fruit peel were changed, and certain sunburn symptoms appeared on the fruit surface. Under the severe sunburn condition, tissue protection ability was reduced, and the cell structure was seriously damaged. In our work, non-bagged fruits did not suffer from sunburn; on the one hand, due to their adaptation to the high-temperature and high-light environment, and on the other hand, as the photoprotection ability and antioxidant enzyme activity of non-bagged fruits were higher than that of debagged fruits, their overall antioxidant capacity was stronger than debagged fruits, and rutin is the most effective to reduce stress among the antioxidants. Therefore, the difference between the severity of oxidative stress and antioxidant capacity determines whether fruit suffers from sunburn.

## Figures and Tables

**Figure 1 molecules-27-03775-f001:**
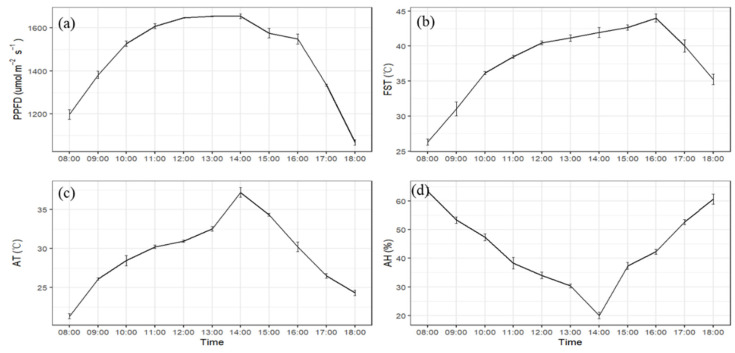
Changes of light intensity, air temperature, humidity, and fruit surface temperature. PPFD: (**a**), photosynthetic photon flux density; FST: (**b**), fruit surface temperature; AT: (**c**), air temperature; AH: (**d**), air humidity. Each index was repeated 5 times.

**Figure 2 molecules-27-03775-f002:**
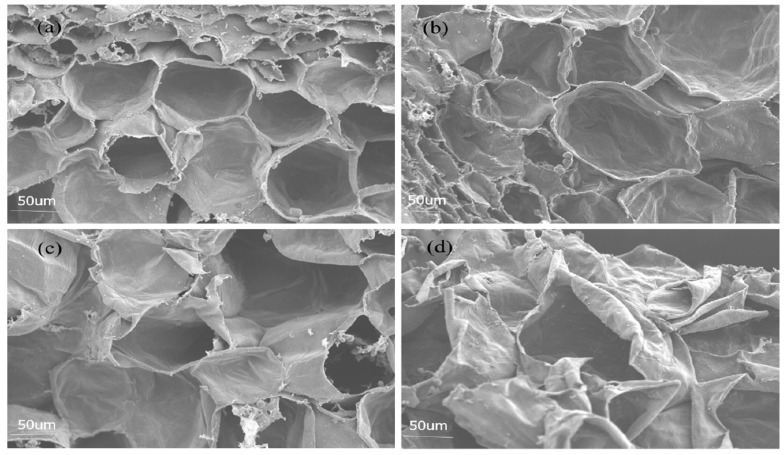
Effects of sunburn on cell structure. (**a**) Debagged fruits, normal (BFN), (**b**) non-bagged fruits (NBF), (**c**) debagged fruits with albefaction (BFA), and (**d**) debagged fruits with browning (BFB).

**Figure 3 molecules-27-03775-f003:**
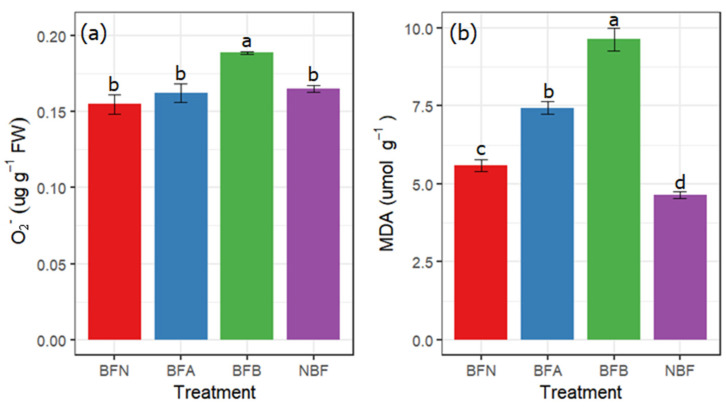
Effects of sunburn on membrane lipid peroxidation. (**a**) Superoxide anion content (O_2_^−^), and (**b**) malondialdehyde content (MDA). Vertical bars are the mean ± SE (*n* = 3). Different letters above the bars indicate significant differences at the 0.05 level by Duncan’s multiple range test.

**Figure 4 molecules-27-03775-f004:**
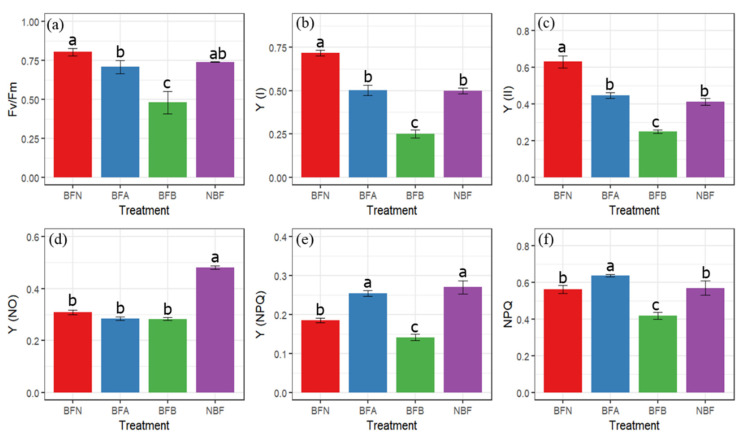
Effects of sunburn on chlorophyll fluorescence parameters. (**a**) *F_v_/F_m_*, (**b**) actual PSI efficiency Y (I), (**c**) actual PSI efficiency Y (II), (**d**) the quantum yield of non-regulated energy dissipation of PSII (Y (NO)), (**e**) the quantum yield of regulated energy dissipation of PSII (Y (NPQ)), and (**f**) non-photochemical quenching (NPQ). Vertical bars are the mean ± SE (*n* = 3). Different letters above the bars indicate significant differences at the 0.05 level by Duncan’s multiple range test.

**Figure 5 molecules-27-03775-f005:**
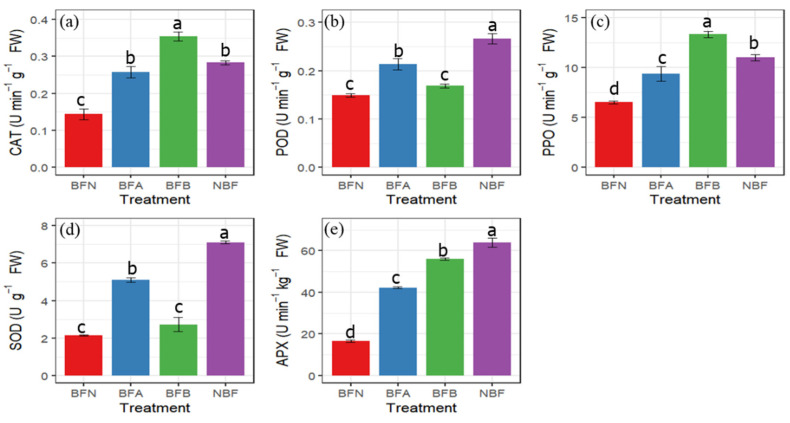
Effects of sunburn on the activities of enzymes. (**a**) Catalase (CAT), (**b**) peroxidase (POD), (**c**) polyphenol oxidase (PPO), (**d**) superoxide dismutase (SOD), and (**e**) ascorbate peroxidase (APX). Vertical bars are the mean ± SE (*n* = 3). Different letters above the bars indicate significant differences at the 0.05 level by Duncan’s multiple range test.

**Figure 6 molecules-27-03775-f006:**
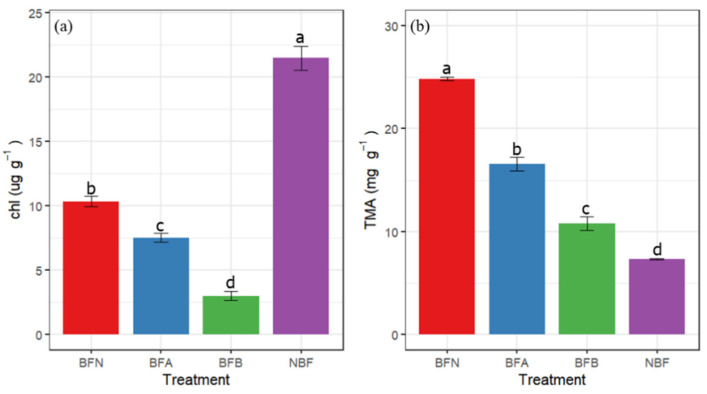
Effects of sunburn on pigment. (**a**) Chlorophyll (Chl), and (**b**) total monomeric anthocyanin (TMA). Vertical bars are the mean ± SE (*n* = 3). Different letters above the bars indicate significant differences at the 0.05 level by Duncan’s multiple range test.

**Figure 7 molecules-27-03775-f007:**
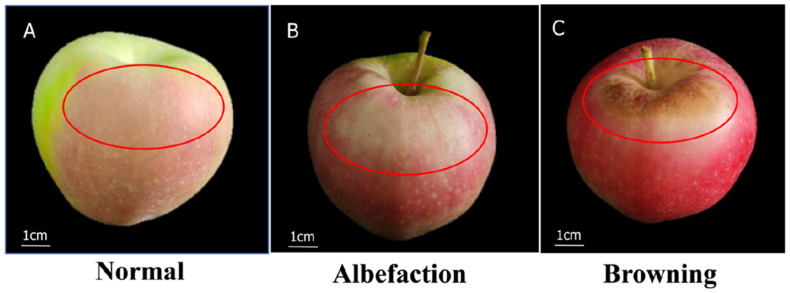
‘Gala’ apples with different levels of sunburn: (**A**) no sunburn, (**B**) photooxidative sunburn, and (**C**) sunburn browning.

**Table 1 molecules-27-03775-t001:** Effects of sunburn on composition of phenolic compounds.

Phenolic Compounds	BFN	BFA	BFB	NBF
Rutin (mg g^−1^)	0.71 ± 0.02 c	5.92 ± 0.12 a	4.07 ± 0.01 b	1.16 ± 0.02 c
Chlorogenic acid (mg g^−1^)	1.24 ± 0.01 ab	1.37 ± 0.01 a	1.44 ± 0.01 a	1.12 ± 0.01 b
Isoquercitrin (mg g^−1^)	0.37 ± 0.01 c	1.06 ± 0.01 a	0.87 ± 0.00 b	0.43 ± 0.00 c
Sum1 (mg g^−1^)	2.32 ± 0.05 c	8.35 ± 0.13 a	6.38 ± 0.03 b	2.71 ± 0.03 c
Procyanidin B2 (mg kg^−1^)	83.75 ± 6.71 b	131.49 ± 9.07 a	71.28 ± 7.76 b	67.73 ± 7.65 b
Epicatechin (mg kg^−1^)	46.88 ± 4.91 c	152.34 ± 6.48 a	91.81 ± 8.24 b	86.36 ± 4.67 b
Catechinic acid (mg kg^−1^)	88.16 ± 3.89 b	133.59 ± 5.19 a	76.23 ± 7.62 bc	54.34 ± 6.48 c
Procyanidin B1 (mg kg^−1^)	40.01 ± 4.85 c	144.40 ± 17.82 a	96.23 ± 6.01 b	75.23 ± 8.40 b
Hyperoside (mg kg^−1^)	97.14 ± 3.82 a	95.58 ± 6.73 a	82.83 ± 4.98 b	72.61 ± 4.98 c
Phloridzin (mg kg^−1^)	4.74 ± 0.35 c	23.29 ± 2.51 a	19.61 ± 1.68 b	4.56 ± 0.85 c
Quercetin (mg kg^−1^)	1.76 ± 0.46 a	0.05 ± 0.01 d	0.17 ± 0.01 c	0.37 ± 0.06 b
Salicylic acid (mg kg^−1^)	0.23 ± 0.03 b	0.21 ± 0.08 b	0.32 ± 0.08 a	0.11 ± 0.01 c
Kaempferol (mg kg^−1^)	0.86 ± 0.21 a	0.69 ± 0.09 b	0.39 ± 0.05 c	0.70 ± 0.32 b
Sum2 (mg kg^−1^)	363.53 ± 24.18 c	681.64 ± 36.31 a	438.87 ± 32.28 b	362.01 ± 32.12 c
Ferulic acid (mg kg^−1^)	5.75 ± 0.33 ab	3.52 ± 0.47 c	7.63 ± 0.48 a	5.11 ± 0.13 b
Methyl gallate (mg kg^−1^)	7.61 ± 1.10 b	13.21 ± 2.29 a	5.28 ± 0.70 c	6.49 ± 0.60 bc
p-Hydroxycinnamic acid (mg kg^−1^)	5.42 ± 0.33 c	11.17 ± 0.79 b	30.15 ± 4.43 a	7.45 ± 0.37 c
Caffeic acid (mg kg^−1^)	25.27 ± 1.42 a	23.20 ± 1.63 a	25.98 ± 1.50 a	19.98 ± 1.92 b
Sum3	44.05 ± 3.15 bc	51.1 ± 5.17 b	69.04 ± 7.09 a	39.03 ± 3.01 c
Sum total (mg g^−1^)	2.68 ± 0.05 c	9.03 ± 0.13 a	6.82 ± 0.03 b	3.07 ± 0.04 c

Notes: The sum of a large number of phenolic compounds is sum1 (mg g^−1^), including rutin, chlorogenic acid, and isoquercitrin. The sum of the medium amount of phenolic compounds is sum2 (mg kg^−1^), from procyanidin B2 to kaempferol. Lower numbers of phenolic compounds is sum3 (ug kg^−1^), from ferulic acid to caffeic acid. The sum of all phenolic compounds is sum total (mg g^−1^). Values are means ± SE values of three replicates. Different letters (a–d) within the same line indicate significant difference at *p* < 0.05 by Duncan’s multiple range test.

## Data Availability

Not applicable.

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
