# Peer review of "Physiological Characteristics of Sunburn Peel after Apple Debagged"

_molecules, 2022, doi:10.3390/molecules27123775_

Round 1
Reviewer 1 Report
i still sustain what i mentionec - there is no validation of quantitative analysis which was done - and in my opinion it is impoertant
- no LOD LOQ caluse
- no intra-inter day values.
- range of concentration of standards etc
- additionally crude extract should be presnted (HPLC) with marked compounds - what you presented it looks like several different extracts, whic is not rue
Reviewer 2 Report
All corrections according to my suggestions and remarks were made.
Author Response
Thank you very much!
Reviewer 3 Report
The paper is focused on the physiological characterization of the apple peel for different treatments. Generally, the paper is well written and deserves to be published. There are some suggestions:
In the introduction, it is necessary to focus more on the literature showing the influence of sunburn on the peel rather than the general facts. For example, “SOD and POD has high infinity for ROS and eliminates hydrogen peroxide in plants [14]”. The reference is about the influence of drought on the leaf of H. vulgare. There are so many studies about fruit peel that could be cited (e.g, The sun-exposed peel of apple fruit has a higher photosynthetic capacity than the shaded peel, Functional Plant Biology 2007, 34(11):1038-1048).
There are important to say which type of sunburn has the apple: sunburn necrosis, sunburn browning, and photooxidative sunburn (see, for example, Sunburn of Apple Fruit: Historical Background, Recent Advances and Future Perspectives, Critical Reviews in Plant Sciences 2012, 31(6):455-504).
Please add the units in table 1.
Author Response
Please see the attachment.

This manuscript is a resubmission of an earlier submission. The following is a list of the peer review reports and author responses from that submission.
Round 1
Reviewer 1 Report
The experiments described has an easy set up, bagged fruit with and without decay were compared with non-bagged ones. It is done statistics on the results, but it is not explained how many replicates and how many fruit in each. Thirty fruit were seleceted, but no further explanation is included. Thirty fruit of each category of one cultivar and one year is a limited material. The introduction do not cite literature about sunburn and it is not possible to know if this work is done before or not. Discussion shows that literature exists. The work should be repeated in order to ensure the results and they should be related to other work in higher degree.
Author Response
Point 1:Thirty fruit were seleceted, but no further explanation is included.
Response 1:Thirty fruits of similar size and location were selected from each group, and each group was treated with three biological replicates, ten fruits per replicate.
Point 2: The introduction do not cite literature about sunburn and it is not possible to know if this work is done before or not. Discussion shows that literature exists.
Response 2:The introduction part has been revised, for the red part of the introduction.
Reviewer 2 Report
molecules-1672686-peer-review-v1
Physiological characteristics of sunburn peel after apple debagged
The manuscript is interesting but needs a lot of improvements. I attached some comments directly to the manuscript
I evaluated mainly phytochemical part of the paper, which is pretty weak. Quantitative analysis should be repeated. There is no information how it was actually done, no validation is presented, how compounds were identified etc. this part of experiments is not well discussed as well

Author Response
Point 1:There is no information how it was actually done, no validation is presented, how compounds were identified etc. this part of experiments is not well discussed as well.
Response 1:Quantitative analysis in 4.8,the specific methods are shown in table 2 and table 3 .
Point 2:I attached some comments directly to the manuscript.
Response 2:It has been revised according to expert advice, in red part of the manuscript.
Reviewer 3 Report
The document is generally well structured. However, there are some sections that could be improved if the terms used are more detailed. English is acceptable, but some grammatical points need to be checked.
The description of the results and their discussion is acceptable.
In the conclusion section, the sentences are not expressed as such. They appear to be a summary of results rather than drawing a conclusion from the data obtained. Conclusions should be specified.

Author Response
Point 1:In the conclusion section, the sentences are not expressed as such. They appear to be a summary of results rather than drawing a conclusion from the data obtained. Conclusions should be specified.
Response 1:It has been revised conclusion section according to expert advice, in red part of the manuscript.
Reviewer 4 Report
Authors propose the results of an experiment aimed to investigate the physiology of the apple fruit peel tissue affected by albefaction and sunburns after debagging. The objective of the research is clear and of scientific relevance. Results showed that in the debagged fruits sunburn, the cell structure of the peel was partially destroyed, the colour of the injured fruit surface turned before white and after brown, and the peroxidation in cell membrane of the peel increased. Initially, the fruit improved its photosynthetic protection ability. The activity of antioxidants and phenolics were enhanced and the increase in membrane peroxidation, resulted in brown coloration of fruits.
The research was correctly designed and carried out and the manuscript is sufficiently well written. In my opinion the manuscript is suitable for publication after a minor revision.
I suggest the authors to provide the following changes.
1) In the results section 2.1, please provide a complete set of air temperature, sun radiation intensity and relative humidity data of the period of fruit development and ripening in the growing locality.
2) Please check your Figure 2 in order to improve the quality of the images. Actually, the hypodermic system is visible only in the Fig. 2a and all the other pictures are limited to the mesenchyme tissue. Please provide images where the epidermis, the hypodermis and the first layers of mesenchyme are clearly visible in order to appreciate the described modifications.
3) Please provide a quantitative indication of the albefaction and sunburn incidence (percentage of affected fruit) respectively in bagged and not bagged fruit. The conclusion affirmation of your abstract “… the non-bagging fruits show higher resistance to high intensity sunlight as compared to debagged fruits …” is not justified by the manuscript results.
4) Please check the Figure 7: change the term “normol” with “normal”.
Author Response
Point 1: In the results section 2.1, please provide a complete set of air temperature, sun radiation intensity and relative humidity data of the period of fruit development and ripening in the growing locality.
Response 1:The author thinks that, the determination of the environmental index of the time from the solution bag to the occurrence of sunburn can be said to be the external conditions of the occurrence of sunburn.
Point 2:Please check your Figure 2 in order to improve the quality of the images. Actually, the hypodermic system is visible only in the Fig. 2a and all the other pictures are limited to the mesenchyme tissue. Please provide images where the epidermis, the hypodermis and the first layers of mesenchyme are clearly visible in order to appreciate the described modifications.
Response 2:The pictures have been replaced according to expert advice.
Point 3:Please provide a quantitative indication of the albefaction and sunburn incidence (percentage of affected fruit) respectively in bagged and not bagged fruit. The conclusion affirmation of your abstract “… the non-bagging fruits show higher resistance to high intensity sunlight as compared to debagged fruits …” is not justified by the manuscript results.
Response 3:It has been revised according to expert advice, in red part of the manuscript.
Point 4:Please check the Figure 7: change the term “normol” with “normal”.
Response 4:It has been revised according to expert advice.
Reviewer 5 Report
Major remarks:
Introduction
- Abbreviations Fv/Fm , SOD and CAT are used for the first time. Explanations are needed.
Results
- If you state “In our work, there was no significant difference in O2- content between BFN and BFA (Fig. 3a).” you need to add NBF too.
- Check is it correct interpretation of Fig 3. “The content of O2− and MDA in NBF were low (Fig.3), which was 0.87 and 0.48 times of BFB respectively.”
- Abbreviations Y(I), Y(NO) and etc are used for the first time. Explanation is needed. Check throughout the text.
- Smaller differences are better to express in percent, not in times, especially if you write “0.28 times”. It is impossible to understand. Correct everywhere in the text, where the difference is less than 100%.
- Statement “The photosynthetic fluorescence parameters of NBF remained at a higher level on the whole (Fig. 4).” is not correct. The same is with BFN.
- To state “POD and SOD initially showed an increasing trend, however, the trend declined with the levels of sunburn increases (Fig. 5b, 5d) is not correct when you have only 3 points.
Materials and methods.
- Indicate country, not only university.
- Instead of M26 write M.26
- Did you cover all fruits on the single tree?
- What you mean by a single replication? “10 apple trees were randomly selected on May 15, 2021 for covering the fruits and other 10 trees as a control where their fruits were not covered in bags in a single replication.”
- 7. mistake of the title of picture A. Should be “Normal.”
- Correct punctuation of following subtitle: 7Chlorophyll. and total monomeric
- “Experimental treatments were arranged as completely randomized design”. But in the beginning you wrote that there was a single replication. It is impossible to randomize single replication.
- Statement is not correct: “(Table 1), The content of phenols in NBF was the lowest, 0.39 mg g-1 lower than that of debagged fruits with normal (BFN)”. It should be vice versa.
Language corrections are needed throughout the text, as for example: “It investigated the atmospheric temperature, humidity, light intensity, and fruit surface temperature of randomly 5 sunny days during the sunburn period “ or “…lower than that of debagged fruits with normal (BFN).” or “According to the method described by Zhang et al. [53] to measure the phenolic compounds (with minor modifications), 3 biological repetitions.”; or “In our work, the early stage of debagged fruit sunburn, the cell structure of the peel was destroyed,..”
The main question is not answered: How severe are sunburn damages in the orchard? What percentage of the yield is affected by sunburn after debagging? Does it cost problems or there are only few cases?
Round 2
Reviewer 1 Report
My last review suggested repetition of the experiment, thats not done.
Reviewer 2 Report
i still sustain what i wrote
there is no information about validation of the method
authors provided only details of LC analysis - no validation is provided
additionally chromatogram should be attached
Reviewer 5 Report
Some work should be done on conclusions.
For example, add sentence from the abstract: " In conclusion, the non-bagging fruits were more adaptable to high intensity sunlight as compared to debagged fruits." Why? Shortly explain the reason.
Write which compounds are the most effective to reduce stress.